# Mechanical Characterization of Thin Asphalt Overlay Mixtures with 100% Recycled Aggregates

**DOI:** 10.3390/ma16010188

**Published:** 2022-12-25

**Authors:** Margherita Pazzini, Giulia Tarsi, Piergiorgio Tataranni, Claudio Lantieri, Giulio Dondi

**Affiliations:** Department of Civil, Chemical, Environmental and Materials Engineering (DICAM), University of Bologna, 40131 Bologna (BO), Italy; margherita.pazzini2@unibo.it (M.P.); piergiorg.tataranni2@unibo.it (P.T.); giulio.dondi@unibo.it (G.D.)

**Keywords:** thin asphalt overlay mixtures, re-surfacing solution, natural and synthetic fibres, recycled aggregates, circular economy

## Abstract

Asphalt pavements inevitably deteriorate over time, requiring frequent maintenance work to ensure the proper serviceability of the road network. Small interventions, such as resurfacing for pavement preservation, are preferable to reconstruction at the end of roads’ in-service lives as they limit environmental- and economic-related impacts. Thin asphalt overlay (TAO) mixture represents a suitable maintenance solution to restore the functional properties of road surfaces. Due to the increasing awareness of the depletion of non-renewable resources and the importance of promoting the circular economy, this study evaluated the possibility of using fully recycled TAO mixes by investigating their volumetric and mechanical properties. Two eco-friendly TAO mixes were designed using recycled aggregates from reclaimed asphalt pavements, a municipal solid waste incinerator, and steel slags in order to meet EN 13108-2 requirements. The TAO mixes differed in regard to the type of bituminous binder (neat/SBS-modified bitumens) and fibres (natural/synthetic) employed. The preliminary results demonstrated that the presence of recycled aggregates did not negatively affect the workability and the mechanical performances of the two sustainable mixtures in terms of stiffness, tensile resistance, rutting and moisture susceptibility. Of these, the TAO mix with neat bitumen and synthetic fibres showed enhanced mechanical performance highlighting the structural effects of the used fibres.

## 1. Introduction

A well-functioning road network is fundamental to the socio-economic and sustainable development of modern societies, playing a crucial role in supporting the growing road traffic, often due to an increase in trade. [1]. At the same time, the regular maintenance of pavements to avoid serious consequences involves continuous investment and improvement works to maintain their operation [2,3]. In order to delay and counteract deterioration, in recent years, asphalt mixtures have been modified and integrated with various materials. Recycled products, fibres, natural and synthetic polymers, fillers and extenders have been added to bituminous binders or into the asphalt mixes during the design of innovative road solutions [4]. For example, treating the aggregate surface with a suitable additive can be a possible solution to enhance moisture damage performance. Nevertheless, these materials may increase the cost of pavement [5]. To satisfy economic purposes as well as the protection of the environment and energy preservation, road pavement researchers and agencies have been looking for alternative materials for the asphalt industry [6,7,8]. 

Waste materials can be used successfully as an aggregate, filler or modifier in asphalt mixtures [9]. Alternatively, they can be used as total or partial bitumen replacement in different layers of pavement structures [10]. Large quantities of waste from commercial, industrial, construction and demolition sectors have been causing serious environmental problems. Replacing raw materials with recycled ones in pavement construction should help waste management and the achievement of sustainability goals in the infrastructure sector [11]. Using waste materials is definitely one of the sustainable strategies for road pavements in urban regeneration [12,13], but this is highly dependent on the selection of appropriate materials in order to keep satisfactory standards [14]. 

Regular surface maintenance before pavement deterioration is always preferred to larger rehabilitation interventions to restore pavement functional conditions [15]. To this purpose, a thin asphalt overlay (TAO) is generally used on existing asphalt concrete pavements in order to restore the surface properties of the layer [16]. As a surface treatment, it provides a skid-resistant finishing, thus reducing the amount of water entering the pavement layers [17]. As a maintenance treatment, TAO is used to fill permanent deformations (ruts) and restore the transverse cross-section profile. The effective service life of TAO varies from 7 to 11 years depending on time of application and the condition of the pavement structure [18]. 

TAO is an asphalt mixture that consists of mineral aggregates and filler in combination with bituminous binders or polymer-modified emulsion, cement and water and often with other additives [19,20]. TAO overlays are typically 20 mm to 30 mm thick [21]. TAO is extremely useful in routine maintenance and pavement preservation and has become a valid solution to restore road pavements as it can improve functional and structural performance in an economical and effective way. This technique is certainly cheaper than laying a thicker layer of dense graded asphalt mixture. A TAO mix is also used for highway maintenance to treat small cracks, road tightness, loss of friction and roughness on a particularly distressed pavement [22].

Given the appropriate conditions, the benefits of placing a TAO mixture are described in various studies [23]. First, it improves surface smoothness, granting better driving conditions, although this depends on the pre-treatment conditions [24]. Second, it reduces wheel path rutting, thus improving safety, provided that the existing pavement rut depth is less than 60 mm [25]. Third, it reduces water intrusion and prevents the loss of structural capacity due to moisture damage. Thin asphalt mixtures, in fact, are impermeable when placed on a compacted asphalt layer with air voids of less than 10% [26]. Through this, they effectively seal all small moderate surface cracks, while larger cracks require further sealing. Fourth, a TAO mix restarts the surface aging process with an increase in the binding properties of the asphalt concrete. When the TAO mixture is placed on existing pavements in good time, the aging process is limited where exposure to heat and oxidation is the highest [27]. Finally, with its macro texture, a new thin asphalt layer reduces the noise generated at the tire–pavement interface [28]. The production of asphalt mixes requires large quantities of natural aggregates whose extraction adversely affects climate change. On the other hand, ever more sustainable alternative solutions are needed to preserve and conserve the environment. For this reason, today, waste materials from the construction and demolition of civil buildings are increasingly being used in road pavements as aggregates in bituminous mixtures in order to reduce environmental impact [29]. Sustainable materials, especially reclaimed asphalt pavements (RAP), reclaimed asphalt shingles (RAS), steel slags, and the residual of industrial and urban waste treatments can also be mixed into TAO mixtures, maintaining surface performance and quality level. The use of RAP in thin asphalt overlay is cheaper and more environmentally viable than a new layer with virgin aggregates [30]. In a different case study, the performances of TAO mixtures containing RAP were even better if compared to reference mixtures produced with virgin aggregates [31]. Steel slag inside TAO can greatly strengthen the skeleton structure and improve its load-bearing capacity. Moreover, after hydrodynamic treatment, the variations in the volumetric parameters and connectivity of TAO with the use of steel slags were shown to be far less significant than the variation in standard overlay [32]. 

Sustainable aggregates can be used inside bituminous mixtures modified with fibres. These mixtures seem to be an excellent solution when applied in TAO mixes on cracked pavements. The high quantity of bitumen in the mastic of these mixtures guarantees a significant resistance to fatigue and reflective cracking. Associated with the stability offered by fibres, bituminous mixtures produce excellent performance, mainly in road pavement overlays [33].

The aim of this study is to design a bituminous mixture, with a low voids content, for a thin hot mix asphalt overlay with 100% recycled aggregates. The use of sustainable materials in TAO mixes proved to be advantageous for the proper maintenance of road pavement surfaces in the presence of minor deteriorations. The novelty of the present study is the use of bituminous mixtures consisting of 100% recycled aggregates, allowing performances comparable to those of traditional hot asphalt mixtures. The recycled aggregates used are RAP, reclaimed municipal solid waste incinerator (MSWI) sand and steel furnace slags. Two types of bitumen were considered: a standard bitumen with a penetration value of 50/70 and an SBS-modified bitumen. In addition, cellulose fibres were added to the mix containing PmB, while innovative synthetic fibres were included in the mixtures with neat bitumen. Thanks to a preliminary mix design, the optimum binder content and the aggregates gradation of the recycled mixtures were defined. The two different TAO mixes were produced and compared in terms of volumetric and mechanical properties in order to evaluate the possible use of these fully recycled mixtures as sustainable maintenance treatment. The experimental programme of this study is visually summarized in Figure 1.

## 2. Materials

Two dense-graded recycled TAO mixtures that differ in the type of bituminous binders and the type and content of fibres were compared in this study. The binder and fibres percentage, together with the aggregates’ type and their grading distribution, were preliminarily defined. The 92%wt. of the lithic skeleton of the TAO mixes consists of recycled aggregates from different origins and the remaining 8% is limestone filler. The grading distribution of aggregates meets the adopted gradation limits for TAO mixture (EN 13108-2). The aggregates gradation is shown in Figure 2. 

The two TAO mixes were prepared using different bituminous binders: a standard 50/70 penetration grade bitumen (PEN 50/70) and a Polymer Modified Bitumen (PmB). The properties of PEN 50/70 and PmB are detailed in a later section. In both mixes, based on preliminary volumetric and mechanical studies, the binder content was fixed at 6.2% by the weight of aggregates. In detail, the air voids content, the stiffness and the tensile resistance of various TAO mixes with different binder content were considered for defining the optimum binder content. The optimum percentage of the binder also counted the aged bitumen already present in the RAP-recycled aggregates.

In the TAO mix with PEN 50/70 (labelled as TAO–PEN 50/70), the 0.5% of aramid-polyolefin fibres (i.e., synthetic fibres) by the weight of aggregates were incorporated during the production. On the other hand, the TAO mixed with PmB (coded as TAO–PmB) included the 0.3% by weight of aggregates of cellulose fibres as stabilizing agent. The fibre content was chosen according to provider information.

### 2.1. Bituminous Binders

The 50/70 penetration grade bitumen (PEN 50/70) and Styrene–Butadiene–Styrene (SBS) polymer modified bitumen (PmB) were used to produce the related TAO–PEN 50/70 and TAO–PmB mixtures, respectively. The properties of the two bituminous binders are listed in Table 1.

### 2.2. Fibres

Cellulose or, alternatively, synthetic fibres that are shown in Figure 3 were used in the present study for TAO mixes. 

The cellulose microfibres have a diameter of 8-20 µm and their length varies between 0.2 and 6.0 mm. The tensile strength of these fibres is declared to vary in the range 1.5–3.0 GPa. This type of fibre is added during the asphalt mixing phase, prior to the addition of bitumen, as a stabilizing agent of the final mixture. Moreover, the presence of cellulose fibres improves the mechanical properties and the skid-resistance finishing of the final asphalt pavement. 

The synthetic macrofibres are a blend of aramid and two types of polyolefin products and are used to improve the final performances of the resulting asphalt mixture. The maximum length of these synthetic fibres is 19 mm. The declared tensile strength of these fibres varies between 70,000 ÷ 400,000 psi (≅0.48 ÷ 2.76 GPa). The aramid component should improve the structural properties of the asphalt concrete, such as cracking and rutting resistance, while the polyolefin fibres are supposed to melt during the mixing process (in fact, the melting point of the synthetic fibres varies between 150 °C and 427 °C based on constituents), modifying the rheological properties of the adopted bitumen. For this reason, a neat bitumen was used for the production of TAO–PEN 50/70 mixture. Additionally, this type of fibre should be incorporated during the asphalt mixing phase, prior to the addition of bitumen.

### 2.3. Aggregates and Filler

The TAO mixes consisted of recycled aggregates of different sizes, which originated from reclaimed asphalt pavement (RAP), municipal solid waste incinerator (MSWI) sand, steel furnace slags and virgin mineral filler.

In the present study, 0–8 mm RAP aggregates with a density of 2.749 g/cm^3^ (EN 12697-6) were used. These recycled aggregates had uniform grading distribution (EN 933-1) and bitumen content equal to 5.2% (EN 12697-1, Soxhelt extraction method) on the total aggregates, which were preliminary evaluated. The bituminous binder that originated from RAP aggregates, albeit aged, can be partially re-activated to offer binding properties; thus, contributing to the cohesion of the final asphalt mixes [34]. The effectiveness of residual binder present in RAP aggregates was evaluated by means of the cohesion test recommended by RILEM [35]. According to the experimental test protocol, the indirect tensile strength (ITS, EN 12697-23) of 100% RAP mixes with no additional virgin bitumen were evaluated at 25 °C after dry and wet conditioning. Samples of RAP mixtures with 150 mm of diameter were produced, and hence compacted, at three different temperatures: 20, 70 and 140 °C. After pre-heating RAP aggregates at each specific production temperature, the 100% RAP mixes were compacted at the three suggested temperatures using a gyratory compactor (30 gyrations, EN 12697-31). The ambient temperature (20 °C) did not allow the compaction of RAP aggregates due to the poor cohesion of the material. Thus, the compacted samples were produced only at 70 and 140 °C. The results of cohesion test on 100% RAP mixtures produced at 70 and 140 °C after dry conditioning for a minimum of 4 h and wet conditioning (in a water bath) for 24 h are listed in Table 2. The results showed the capacity of RAP binder to actively contribute to the cohesive properties of the asphalt mixture when heated, since their values exceeded the lower ITS limit (i.e., 0.10 MPa) suggested by the RILEM testing protocol in all conditions. 

The steel slag aggregates with a diameter in the range of 4–8 mm were used. Due to the presence of iron oxides, this type of aggregate has a higher density than mineral ones, which was equal to 3.856 g/cm^3^ (EN 12697-6). These aggregates were subjected to preliminary geometrical and physical characterizations, showing a flakiness index (FI, EN 933-4)) of 2.4%, a resistance to fragmentation of 13% (EN 1097-2) and a high resistance to freezing and thawing without weight loss (EN 1367-1). These recycled aggregates showed geometrical and physical properties similar to virgin aggregates’ increased resistance to fragmentation, confirming previous studies [36,37]. These characteristics are the basis of the high microtextural properties of recycled aggregates that make them widely used in pavements subjected to significant traffic loads and tangential stresses. In addition, the affinity between steel slags and the two bituminous binders was investigated through the rolling bottle test (EN 12697-11). The degree of coverage of each binder on the aggregates was analysed after 6, 24 and 48 h and the results are listed in Table 3. The PmB showed a greater affinity with the aggregates compared to the neat bitumen in all testing intervals, which resulted in increased stripping resistance. Moreover, the rate of coverage degradation of steel slags–PmB was lower than the compared steel slags–PEN 50/70.

Recycled sand coming from the MSWI was used for the production of recycled TAO mixtures. Aggregates with diameter varying in the range of 0–5 mm were used. The density of this material was 2.564 g/cm^3^ (EN 12697-6). The sand equivalent value (EN 933-8) of the 0–2 mm fraction of this fine aggregate was determined, which resulted in an HS value of 76%.

Traditional limestone filler with a diameter lower than 0.063 mm was introduced to complete the aggregate gradation of the TAO mixes, thus meeting the specification of the EN 13108-2 standard. 

Based on the size of recycled aggregates, the composition of the lithic skeleton of TAO mixtures was:

Steel slag aggregates: 51%;RAP aggregates: 20%;Artificial sand from MSWI: 21%;Filler: 8%.

## 3. Methods

The two recycled TAO mixtures, namely TAO–PEN 50/70 and TAO–PmB, were prepared in order to evaluate their volumetric and mechanical properties, such as tensile resistance, stiffness, moisture and rutting susceptibility. The experimental programme included the following characterizations:

air voids content, EN 12697-8;indirect tensile strength (ITS), EN 12697-23;indirect tensile stiffness modulus (ITSM), EN 12697-26;indirect tensile strength ratio (ITSR), EN 12697-12;Hamburg wheel track test, AASHTO T 324-11.

A total of 19 samples with a maximum thickness of approximately 3.8 mm were produced at about 170 °C for each mix. The samples thickness met the upper limit indicated by the Federal Highway Administration (FHWA) for TAO mixtures [38] allowing all mechanical tests to be performed without overlaying the sample on a bottom asphalt mixture. The samples with a diameter of 150 mm were compacted by means of a gyratory compactor according to the EN 12697-31. Depending on the type of testing, TAO specimens were compacted with different numbers of gyrations. 

The compaction curves obtained from the gyratory compactor were analysed in order to evaluate the volumetric properties of the experimental mixtures. The air voids (Va) content of the samples was measured (EN 12697-8) at three specific number of gyrations, namely 20, 120 and 180.

In terms of mechanical properties, the cohesion of the asphalt mixes was investigated by means of indirect tensile strength (ITS) test according to the EN 12697-23 standard. During this static mechanical analysis, the specimen was subjected to a load with a constant velocity of 50 mm/min until break. The test temperature was fixed at 25 °C. For this reason, all samples were thermally conditioned for 4 hours before being tested.

The stiffness of TAO mixes was determined using the indirect tensile stiffness modulus (ITSM) test according to the EN 12697-26 standard in the indirect tensile configuration (IT-CY, Annex C of the standard). As indicated in the standard, a pulse loading with a rise time of 124 ms generates a predefined horizontal deformation of 5±2 μm, allowing the evaluation of the ITSM value of an asphalt mixture. This test was carried out at three test temperatures: 10, 20 and 40 °C.

The samples used for volumetric, static and dynamic mechanical characterizations were compacted applying 180 revolutions.

The moisture susceptibility of the two recycled TAO mixes was investigated by the use of indirect tensile strength ratio (ITSR) according to Method A of the EN 12697-12 standard. The ITSR value represents the ratio between the average ITS value obtained from samples conditioned in air (dry conditioning) and those conditioned in a water bath (wet conditioning). The wet conditioning involved the storage of specimens in a water bath at 40 °C for 72 h. The samples were compacted by applying 50 revolutions.

The rutting resistance and water damage of the recycled TAO mixtures were further estimated through the Hamburg wheel track test according to the AASHTO T 324-11 standard. The device simulates the traffic loads that pavements experience while in service. During the laboratory test, the deterioration of samples was accelerated by applying the load over samples in a water bath at 50 °C. A steel wheel passed over two samples in a specific configuration according to 52 ± 2 passes per minute with a maximum speed of approximately 0.305 m/s. The test measurement corresponds to the deformation (rut) caused by the wheel.

## 4. Results and Discussion

### 4.1. Volumetric Characterization

The volumetric characterization allowed the evaluation of the compaction properties and workability of the two experimental mixtures produced with recycled materials. Figure 4 shows the average compaction curves for the two TAO mixes. 

The curves were defined based on the progressive density of samples calculated after each revolution of the gyratory compactor. Data were represented with the following Equation (1): 
(1)
$$\%\rho_{m}=a\cdot \ln\left(x\right)+b$$

where %*ρ_m_* is the percentage of maximum densification, *a* is the slope of the compaction curve, *x* is the number of gyrations and *b* is the intercept of the regression curve. Figure 4 shows the compaction curves of the two recycled TAO mixtures, while Table 4 lists the coefficients used in the Equation (1) and the air voids contents (Va) at three different numbers of gyrations. 

The compaction curves, model coefficients and the air voids contents of the two TAO mixtures were similar, showing no significant differences in terms of the maximum density achieved at the end of the compaction process (i.e., 180 gyrations) and, hence, the workability. Although similar and in line with current specifications, the air voids of TAO–PEN 50/70 were lower than those of TAO–PmB at all progressive gyration numbers. The major air voids difference between the designed mixes was observed at the end of the compaction, where the values varied by about 1%. The increased density of this mixture could be attributed to the presence of aramid–polyolefin fibres, which might reduce the porosity of the asphalt concrete. 

### 4.2. Static Mechanical Analysis: ITS Results

The indirect tensile strength (ITS) test was used to evaluate the tenacity of the aggregates–filler–bitumen bond of the recycled TAO mixes. Three samples per each mixture were conditioned in a climate chamber before being tested and the average ITS values of the two TAO mixtures are listed in Table 5. Both mixes met the reference Italian specification (i.e., 1.0 MPa < ITS < 1.80 MPa). The high percentage of RAP aggregates led to the enhancement of the tensile resistance of the two TAO mixes. However, between the two analysed mixtures, the TAO–PEN 50/70 with synthetic fibres showed a higher ITS value than the mix produced with PmB and cellulose fibres. Based on the obtained data, the contribution of the synthetic fibres in enhancing the cohesion of the asphalt concrete was evident. Thus, the combination of these fibres with a neat bitumen allowed the achievement of higher tensile resistance even when compared to a bituminous mixture produced with PmB. However, it is desirable that the ITS values of TAO mixes do not overly exceed the upper limit to avoid the production of a very stiff, and therefore brittle, asphalt concrete. 

### 4.3. Dynamic Mechanical Analysis: ITSM Results

The dynamic mechanical characterization was performed on three samples per each TAO mix at three different test temperatures in order to evaluate the thermal susceptibility of the asphalt concrete. The average ITSM values are reported in Figure 5. 

Figure 5 clearly shows the higher stiffness of the mixture produced with neat bitumen and synthetic fibres at every test temperature. In accordance with the previous static mechanical analysis, the incorporation of aramid–polyolefin fibres allowed the enhancement of the asphalt properties under every condition. This further confirmed the achievement of high mechanical properties with the combined use of neat bitumen and synthetic fibres. The obtained ITSM values for TAO–PEN 50/70 were comparable and even higher than the stiffness of the TAO–PmB. Overall, the ITSM values of both mixtures satisfied the referenced Italian technical specifications. 

In general, the stiffness of the designed mixtures was higher than standard TAO mixtures and dense-graded wearing course layers, and this behaviour can be attributed to the presence of RAP aggregates. In particular, at 40 °C, both TAO mixes had a significant stiffness showing no particular decay in their mechanical properties at high temperatures, which might be favoured by the reduced thickness of the asphalt layer. 

As for the thermal susceptibility, there was no considerable difference between the trends shown by the two mixtures. This phenomenon is significant, considering the different rheological properties of the adopted bituminous binders. 

### 4.4. Moisture Susceptibility

The moisture susceptibility of the TAO mixes was evaluated by comparing the ITS values of dry and wet conditioned samples. Six samples per each TAO mix were subjected to conditioning and ITS testing according to the EN 12697-12 standard. The average ITS results of the recycled TAO mixes after dry and wet conditioning are reported in Figure 6, which includes the corresponding ITSR results.

The trend of the tensile resistance of the two recycled TAO mixes was further confirmed: TAO–PEN 50/70 achieved higher ITS values, both in dry and wet conditions. Due to the lower compaction (i.e., lower number of revolutions) during the production of these specimens, the resulting ITS values were lower than those listed in Table 4. As for the TAO–PEN 50/70 mix, the ITS value increased after being conditioned in a water bath. This phenomenon could be related to the presence of RAP within the mixture [39]. On the other hand, the wet conditioning slightly decreased the tensile strength of the TAO–PmB samples since the resulting ITSR value was found to be equal to 98%. Both recycled TAO mixes experienced no significant reduction in ITS values after the water conditioning, being far from 75%, which is indicated as the lower ITSR limit in the relevant Italian technical specifications. In addition, both TAO mixes complied with the EN 13108-2 standard limits. However, the TAO mix produced with PEN 50/70 and aramid–polyolefin fibres showed a reduced moisture susceptibility compared to the recycled TAO–PmB mix, which may be attribute to the different bituminous binder and fibres used. In general, PmBs are expected to be less prone to deterioration under moisture/water exposition, while cellulose fibres may speed up the diffusion of moisture into the asphalt concrete. The combination of the two behaviours could be blamed for the almost unchanged ITSR value of the TAO–PmB.

### 4.5. Rutting Resistance

The rutting resistance of the recycled TAO mixtures was evaluated according to the AASHTO T 324-11 standard. The objective of this test is the simulation of an accelerated rutting deformation on the asphalt concrete caused by the passing of a steel wheel. When the test is carried out in wet conditions, the results can also be related to the water damage resistance of the asphalt concrete. In the case under study, the test was performed in a water bath at 50 °C and the failure criteria considered the achievement of a 20 mm rut depth or 20,000 passes. The development of the rut depth for the two mixtures is represented in Figure 7.

In both cases, the samples reached 20,000 passes, showing considerable structural properties. Overall, the recorded rut depth of the TAO PEN 50/70 was lower (1.71 mm) when compared to the value reached by TAO–PmB (2.52 mm). These data are in line with the previous tests and validate the function of the synthetic fibres in improving the performance of the asphalt concrete. Furthermore, it is worth noting that the test temperature was the same as that indicated as the softening point temperature for the neat bitumen. Thus, the good rutting resistance achieved for TAO–PEN 50/70 might be also related to the possible modification of the rheological properties of the binder conferred by the addition of the aramid–polyolefin fibres.

Looking at the curves, the trends of the rut depth of both TAO mixtures were similar at the beginning of the test, showing a comparable post-consolidation phase until around 1000 passes. In theory, after the post-compaction consolidation the curve could be divided in two consecutive sections: the creep slope and the stripping slope. The first is the inverse of the rutting slope and it is used to evaluate the rutting potential of the material. The second is a measurement of the accumulation of moisture damage. The intercept of the creep and stripping slope is called stripping inflection point (SIP), and it is generally used to evaluate the moisture damage potential of the asphalt concrete. In the case under study, the TAO–PEN 50/70 showed an almost flat trend after the post-compaction consolidation, without reaching the SIP after 20,000 passes. As for the TAO–PmB mixture, the SIP was reached after 12,438 passes. Although the latter mixture showed a lower susceptibility to water damage, it was evident that the use of synthetic fibres improved rutting resistance. 

## 5. Conclusions

In the present study, a laboratory characterization of two fully recycled thin asphalt overlay (TAO) mixtures is presented. The two eco-friendly asphalt mixes consisted of 100% recycled aggregates, differing in the types of bituminous binder and fibres used. Aramid–polyolefin fibres were added to the mix with a standard 50/70 penetration grade bitumen, while cellulose fibres were incorporated in the production of the mixture with SBS-modified bitumen. The experimental programme preliminarily evaluated and compare the volumetric and mechanical properties of two sustainable TAO mixes, while considering the European and Italian specifications as a reference. On the basis of the experimental data presented in this paper, the following conclusions can be drawn:

Despite the prominent presence of recycled aggregates (RAP, steel slags and sand from MSWI), the designed mixture met the gradation limits imposed. Moreover, the workability and compactability properties of both TAO mixes were not negatively affected, as verified in the volumetric analysis.Both TAO mixtures complied with the requirements indicated in the EN 13108-2 standard and in the adopted Italian technical specifications in terms of air voids, indirect tensile strength, stiffness and moisture damage resistance. Overall, the massive use of recycled aggregates did not imply a detrimental effect in the physico-mechanical performance of the final asphalt concrete.The higher mechanical properties of TAO–PEN 50/70 mixture were clearly seen in each test. The aramid–polyolefin fibres contributed to improve the cohesion of the mixture, obtaining higher ITS values in both dry and wet conditions (higher ITSR values) and its structural properties enhancing both its ITSM values at each test temperature and its rutting resistance. Furthermore, the addition of these synthetic fibres together with neat bitumen reduced the water susceptibility of the final TAO mixture.The combined use of aramid–polyolefin fibres and neat bitumen allowed the achievement of similar performance of the TAO mixture produced with PmB and natural fibres. No significant differences were verified in terms of thermal susceptibility between the two analysed TAO mixes. Considering that this property is traditionally linked to the rheological characteristics of the adopted bitumen, a modification in these parameters might have occurred when melting the aramid–polyolefin fibres with the PEN 50/70. 

Overall, the designed mixtures seem to be a suitable and sustainable alternative to traditional hot asphalt mixtures for thin-wearing course overlays, allowing the massive use of recycled materials without negatively affecting the final performances of road pavements.

Future specific tests will be necessary to completely characterize the proposed recycled TAO mixtures. In detail, the adhesion between the TAO mixes and traditional asphalt concrete during their overlay would be crucial for assessing the feasible use of the proposed mix for road surface maintenance. In fact, the presence of fibres may influence the adhesion between different asphalt layers. Moreover, the effective changes in the rheological properties of the neat bitumen when the aramid–polyolefin fibres are incorporated will require further investigations. Thanks to the promising preliminary results, the sustainability of the proposed asphalt mixes will be assessed from an environmental and economic point of view when performing life cycle–cost analyses.

## Figures and Tables

**Figure 1 materials-16-00188-f001:**
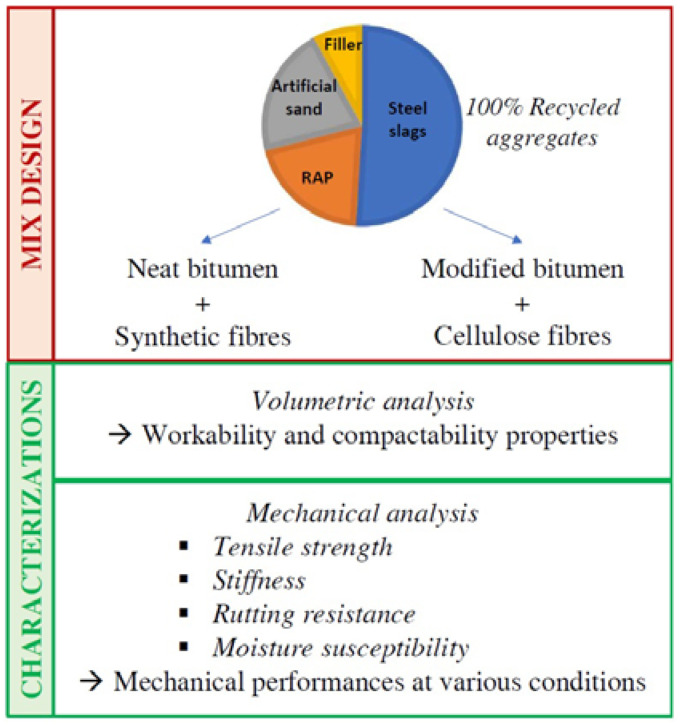
Summary of the experimental programme.

**Figure 2 materials-16-00188-f002:**
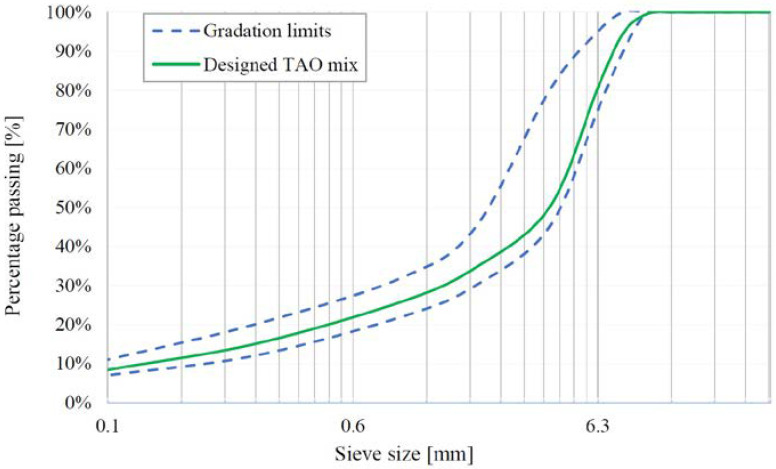
Aggregates distribution and gradation limits.

**Figure 3 materials-16-00188-f003:**
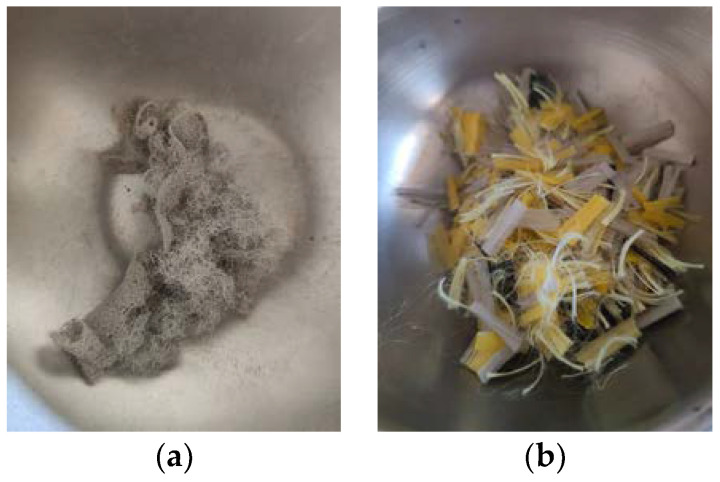
Stabilizing agents of TAO mixes: (**a**) cellulose fibres; (**b**) synthetic fibres.

**Figure 4 materials-16-00188-f004:**
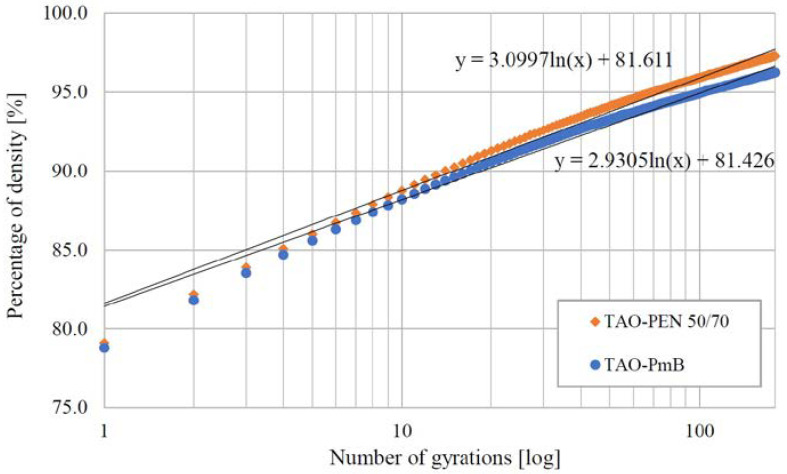
Compaction curves of TAO–PEN 50/70 and TAO–PmB and their models.

**Figure 5 materials-16-00188-f005:**
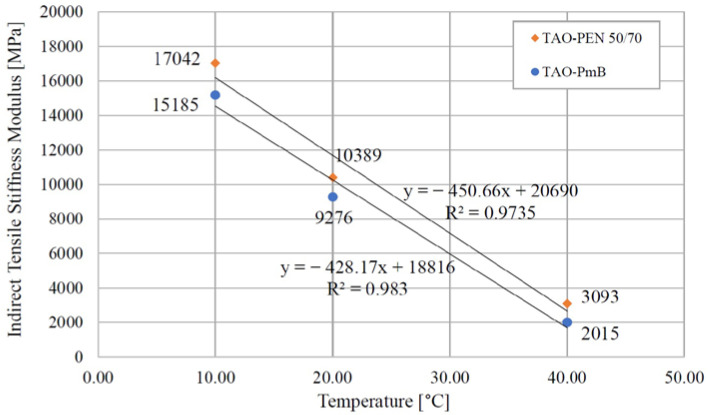
Average ITSM values of TAO–PEN 50/70 and TAO–PmB.

**Figure 6 materials-16-00188-f006:**
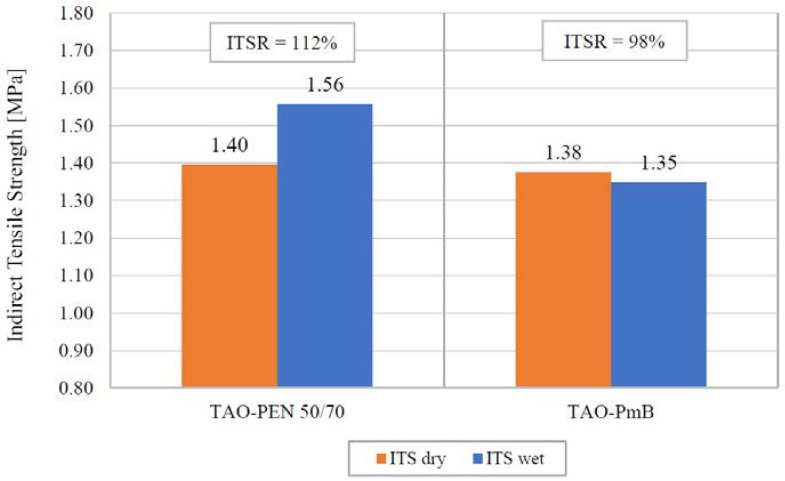
Average ITSR data from the ITS values at dry and wet conditioning of TAO–PEN 50/70 and TAO–PmB.

**Figure 7 materials-16-00188-f007:**
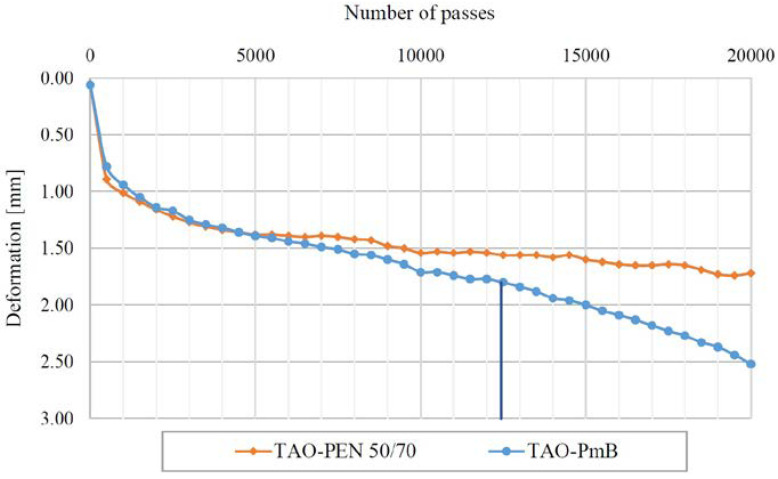
Rut depth of TAO–PEN 50/70 and TAO–PmB obtained from Hamburg wheel track test.

**Table 1 materials-16-00188-t001:** Technical properties of PEN 50/70 and PmB.

Bituminous Binder	Penetration at 25 °C (EN 1426)	Scheme 1427.	Resistance to Short Term Aging: RTFOT (EN 12607-1)	
			Mass Variation	Residual Penetration
PEN 50/70	54 dmm	46.9 °C	0.5%	50%
PmB	45 dmm	88.0 °C	0.1%	65%

**Table 2 materials-16-00188-t002:** ITS values of 100% RAP mixes produced at 70 and 140 °C after dry/wet conditioning.

Production Temperature [°C]	Type of Conditioning	Average ITS [MPa]
70	dry	0.25 ± 0.02
	wet	0.11 ± 0.02
140	dry	1.35 ± 0.13
	wet	0.82 ± 0.26

**Table 3 materials-16-00188-t003:** Results of rolling bottle test on steel slag aggregates and the considered bituminous binders.

Time Interval [h]	Degree of Coverage		Rate of Degradation	
	PEN 50/70	PmB	PEN 50/70	PmB
0 (Start test)	100%	100%	-	-
6	60%	75%	−40%	−26%
24	35%	57%	−41%	−23%
48	25%	44%	−30%	−23%

**Table 4 materials-16-00188-t004:** Parameters of compaction curves models and air voids of TAO–PEN 50/70 and TAO–PmB.

Mixture	a	b	Air Voids, Va		
			10 Gyrations	120 Gyrations	180 Gyrations
TAO–PEN 50/70	3.100	81.611	11.3%	3.7%	2.7%
TAO–PmB	2.931	81.426	11.7%	4.5%	3.6%

**Table 5 materials-16-00188-t005:** Average ITS values of TAO–PEN 50/70 and TAO–PmB.

Mixture	ITS
	[MPa]
TAO–PEN 50/70	1.78 ± 0.05
TAO–PmB	1.58 ± 0.06

## Data Availability

The data presented in this study is available upon request from the corresponding author.

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
