# Peer review of "Mechanical Characterization of Thin Asphalt Overlay Mixtures with 100% Recycled Aggregates"

_materials, 2022, doi:10.3390/ma16010188_

Round 1

Reviewer 1 Report

The paper is in general terms well written, and it is referred to an interesting subject as it is the use of recycled aggregates in the design of Thin Asphalt mixtures. Nonetheless, below are few comments that will help the authors in enhancing the quality of the same.

 -        Line 37. It would be suitable to include some more references as example of the use of these kinds of materials. Some examples can be found in:

Tauste-Martínez, R., Moreno-Navarro, F., Sol-Sánchez, M., & Rubio-Gámez, M. C. (2021). Multiscale evaluation of the effect of recycled polymers on the long-term performance of bituminous materials. Road Materials and Pavement Design, 22(sup1), S99-S116.

Hidalgo, A. E., Moreno-Navarro, F., Tauste, R., & Rubio-Gámez, M. C. (2020). The influence of reclaimed asphalt pavement on the mechanical performance of bituminous mixtures. An analysis at the mortar scale. Sustainability, 12(20), 8343.

-        Line 112. Despite the design of the two proposed mixtures, it would be necessary to include a third mixture of reference so to fully assess the benefits of the new mixtures proposed when comparing to a solution that already is employed.

-        Lines 122-123. Include the reference to these previous studies.

-        Line 317. Although authors include a reference so to explain the higher values of wet group in relation to the dry one, a more extend explanation is required focusing on the reason behind this phenomenon does not occur in the mixture manufacture with modified bitumen.

Author Response

We would like to thank the reviewer for the comments and suggestions on our contribution ‘materials-2084323’ submitted for the Journal of Materials – Section: Construction and Building Materials – Special Issue: Recycling Pavements Materials. We have carefully taken the reviewer’s comments into consideration and addressing them all in the revised manuscript. Hereafter we provide a point-by-point response.

With best regards on behalf of all authors,

Giulia Tarsi.

The paper is in general terms well written, and it is referred to an interesting subject as it is the use of recycled aggregates in the design of Thin Asphalt mixtures. Nonetheless, below are few comments that will help the authors in enhancing the quality of the same.

Point 1: Line 37. It would be suitable to include some more references as example of the use of these kinds of materials. Some examples can be found in:

  • Tauste-Martínez, R., Moreno-Navarro, F., Sol-Sánchez, M., & Rubio-Gámez, M. C. (2021). Multiscale evaluation of the effect of recycled polymers on the long-term performance of bituminous materials. Road Materials and Pavement Design, 22(sup1), S99-S116.
  • Hidalgo, A. E., Moreno-Navarro, F., Tauste, R., & Rubio-Gámez, M. C. (2020). The influence of reclaimed asphalt pavement on the mechanical performance of bituminous mixtures. An analysis at the mortar scale. Sustainability, 12(20), 8343.

Response 1: The authors thank you for the suggestion. The literature has been improved according to the references recommended.

Point 2: Line 112. Despite the design of the two proposed mixtures, it would be necessary to include a third mixture of reference so to fully assess the benefits of the new mixtures proposed when comparing to a solution that already is employed.

Response 2: The authors have considered the European standard requirements as a reference to assess the feasibility of using an eco-friendly TAO mixture that contains recycled aggregates only. Since the study needs further investigations, the design and characterization of an additional TAO mix with virgin aggregates only will be also considered.

Point 3: Lines 122-123. Include the reference to these previous studies.

Response 3: Thanks to preliminary analysis on this specific study the optimum binder content has been defined for both TAO mixtures. In order to clarify this point, the word ‘previous’ was substituted with ‘preliminary’ and the performed tests were added to the description. Please, refer to lines 138-139.

Point 4: Line 317. Although authors include a reference so to explain the higher values of wet group in relation to the dry one, a more extend explanation is required focusing on the reason behind this phenomenon does not occur in the mixture manufacture with modified bitumen.

Response 4: Based on your comment the description of paragraph 3.4 has been improved accordingly. The almost unchanged ITSR value of TAO-PmB rather than the higher ITSR value of TAO-PEN 50/70 may be related to the different behaviour of the polymer-modified bitumen in combination of the cellulose fibres. PmBs are expected to be less prone to water/moisture susceptibility than standard bitumen, while cellulose fibres may speed up the diffusion process into the asphalt concrete. Please, refer to lines 368-372 for manuscript changes.

Reviewer 2 Report

Dear Authors

Thanks for your effort, it's a good article about the use of fibers and recycled asphalt in the flexible pavement, but some points of ambiguity and controversy will be raised in the following:

-The specifications of these two materials should be given in the text of the article.

a) cellulose fibers; b) synthetic fibers.

-What is the meaning of circular economy[Keywords]? In this article, in which part is it used specifically?

-Pictures of asphalt samples should be added to the article, how did you apply Thin Asphalt Overlay on the asphalt sample? Please give an explanation.

-In the abstract, the numerical results of this study should also be mentioned.

-Give the specifications of the recycled asphalt used in this article. Were these materials uniform in terms of granulation and the amount of bitumen available? Were the characteristics of the aged bitumen determined in this recycled material? please explain.

-The adhesion of the Thin Asphalt Overlay and the asphalt in the bottom layer changes when the fiber is used (it acts as a barrier), how is this function considered in this study?

Thanks

Author Response

We would like to thank the reviewer for the comments and suggestions on our contribution ‘materials-2084323’ submitted for the Journal of Materials – Section: Construction and Building Materials – Special Issue: Recycling Pavements Materials. We have carefully taken the reviewer’s comments into consideration and addressing them all in the revised manuscript. Hereafter we provide a point-by-point response.

With best regards on behalf of all authors,

Giulia Tarsi.

Dear Authors, thanks for your effort, it's a good article about the use of fibers and recycled asphalt in the flexible pavement, but some points of ambiguity and controversy will be raised in the following:

Point 1: The specifications of these two materials should be given in the text of the article.

  1. cellulose fibers; b) synthetic fibers.

Response 1: The main specifications of the two fibres used were reported in paragraph 2.2, where additional information was provided to the readers while revising the manuscript. In detail, please, refer to lines 163-164 and 170-171.

Point 2: What is the meaning of circular economy [Keywords]? In this article, in which part is it used specifically?

Response 2: The keyword ‘circular economy’ has been added to highlight the use of recycled aggregates. In the present study, recycled aggregates from wastes and/or by-products of different industrial chains have been used to improve the sustainability of the final asphalt mixes. The authors have changed the order of keywords in order to better connect the terms ‘circular economy’ and ‘recycled aggregates’.

Point 3: Pictures of asphalt samples should be added to the article, how did you apply Thin Asphalt Overlay on the asphalt sample? Please give an explanation.

Response 3: Unfortunately, the authors did not take pictures of samples unless they were under testing devices. However, the TAO mixes were not overlayed on a bottom asphalt sample. This study wanted to represent a preliminary investigation on the characteristics of recycled TAO mixtures producing samples 3.8 mm approx. thick. The considered thickness allows the performance of the planned volumetric and mechanical tests. The authors referred to the document of the Federal Highway Administration, FHWA-HIF-19-053, entitled ‘The use of Thin Asphalt Overlay for Pavement Preservation’ to consider the thickness of the asphalt mixtures’ samples. The authors thank you for the comment and in order to clarify this aspect some changes have been introduced in the description of ‘Methods’ section, please see lines 247-251.

Point 4: In the abstract, the numerical results of this study should also be mentioned.

Response 4: The abstract has been improved trying to better highlight the results and conclusions obtained from the experimental analysis.

Point 5: Give the specifications of the recycled asphalt used in this article. Were these materials uniform in terms of granulation and the amount of bitumen available? Were the characteristics of the aged bitumen determined in this recycled material? please explain.

Response 5: The authors evaluated the grading distribution and the bitumen content of the used RAP aggregates. In order to clarify this aspect, lines 184-187 of the manuscript were changed accordingly. The cohesive property of the aged bitumen present in RAP aggregates was investigated following the RILEM recommendation as explained in lines 189-197; while, the physico-rheological properties of the extracted bitumen from RAP were not analysed. The characteristics of aged bitumen and the interaction between this binder and the virgin ones can be further investigated.

Point 6: The adhesion of the Thin Asphalt Overlay and the asphalt in the bottom layer changes when the fiber is used (it acts as a barrier), how is this function considered in this study?

Response 6: This study did not evaluate the adhesion of the Thin Asphalt Overlay mixtures and the bottom layer as it wanted to represent a preliminary study on the characterisation of recycled TAO mixtures. In detail, all asphalt samples were produced and tested without overlayed them on a bottom sample since the maximum thickness of the TAO mix is 3.8 cm approximately. The considered thickness allows the performance of all planned volumetric and mechanical tests. The authors referred to the document of the Federal Highway Administration, FHWA-HIF-19-053, entitled ‘The use of Thin Asphalt Overlay for Pavement Preservation’ to consider the thickness of the asphalt mixtures’ samples. The effect of fibres on the final adhesion of the asphalt layer on a bottom one will be further investigated following your suggestion. In order to clarify this aspect, some changes have been introduced in the description of ‘Methods’ and in the ‘Conclusions’ section, please refer to lines 247-251 and 441-444, respectively.

Reviewer 3 Report

The manuscript entitled “Laboratory characterization of sustainable Thin Asphalt Overlay mixes with recycled aggregates” presented experimental and statistical study. The influence of different parameters was studied and analyzed.

This reviewer recommends major editing and resubmits for re-review.

Comments:

  • The English writing of the manuscript needs improvement. Therefore, it could benefit greatly from professional editing to improve technical writing and English.
  • Please mention your study limits and suggest some future research topics
  • In References, the sources are written in different styles. Please update the reference list.  It is necessary to bring in accordance with the requirements of the magazine for the design of References. If possible, indicate DOI.
  • The literature can be expanded by studying some of these papers.
    • Predicting Marshall Flow and Marshall Stability of Asphalt Pavements Using Multi Expression Programming
    • Evaluation of Moisture Damage Potential in Hot Mix Asphalt Using Polymeric Aggregate Treatment
  • Please use some innovative keywords.
  • Please mention your study limits in the abstract.
  • The Conclusions should reflect what the practical application of the results obtained in this study is. In what climatic conditions should the recommendations of the authors be taken into account?
  • The authors should increase their discussion on previous related research and highlight how their study is providing a different approach or adding significantly to what has been done. The authors have to explain what is the new here in comparison with the previous studies. The novelty of the current work should be highlighted in the introduction. Please try to mention a problem that needs solving - in other words, the research question underlying your study clearer.
  • The title of the manuscript should be revised.
  • Some types of standards should be used to perform different experimental studies. Please provide details for the standards used in each study.
  • Section 4 should be discussed in detail.
  • The authors must redo the Abstract and bring it in compliance with the requirements of the journal. The scientific problem is poorly described (Background). The scientific novelty is not indicated. I recommend shortening the Abstract to 200 words. Editors strongly encourage authors to use the following style of structured abstracts, but without headings: (1) Background: Place the question addressed in a broad context and highlight the purpose of the study; (2) Methods: Briefly describe the main methods or treatments applied; (3) Results: Summarize the article's main findings; and (4) Conclusions: Indicate the main conclusions or interpretations. The abstract should be an objective representation of the article
  • It is advisable to add a flowchart at the beginning of the paper. Then the article would become more visual and structured
  • The economic aspects are also required for sustainability in social aspect. It is suggested to authors to evaluate the cost-benefit study of this as a further investigation
  • The conclusion should be an objective summary of the most important findings in response to the specific research question or hypothesis. A good conclusion states the principal topic, key arguments and counterpoint, and might suggest future research. It is important to understand the methodological robustness of your study design and report your findings accordingly. Please improve your conclusion section.

Author Response

We would like to thank the reviewer for the comments and suggestions on our contribution ‘materials-2084323’ submitted for the Journal of Materials – Section: Construction and Building Materials – Special Issue: Recycling Pavements Materials. We have carefully taken the reviewer’s comments into consideration and addressing them all in the revised manuscript. Hereafter we provide a point-by-point response.

With best regards on behalf of all authors,

Giulia Tarsi.

The manuscript entitled “Laboratory characterization of sustainable Thin Asphalt Overlay mixes with recycled aggregates” presented experimental and statistical study. The influence of different parameters was studied and analyzed.

This reviewer recommends major editing and resubmits for re-review.

Point 1: The English writing of the manuscript needs improvement. Therefore, it could benefit greatly from professional editing to improve technical writing and English.

Response 1: The authors have improved the English writing of the manuscript following your comment.

Point 2: Please mention your study limits and suggest some future research topics

Response 2: Thank you for your suggestion. Based on your comments, the authors improved paragraph 4 (Conclusions) adding the study limits and the advisable/necessary further investigations to complete this preliminary experimental study. Please, refer to ‘Conclusions’ section, especially from line 440, for all detailed information.

Point 3: In References, the sources are written in different styles. Please update the reference list.  It is necessary to bring in accordance with the requirements of the magazine for the design of References. If possible, indicate DOI.

Response 3: The authors thank you for noting the inaccuracy and underlining it. The references have been modified according to the guidelines of the journal.

Point 4: The literature can be expanded by studying some of these papers.

  • Predicting Marshall Flow and Marshall Stability of Asphalt Pavements Using Multi Expression Programming
  • Evaluation of Moisture Damage Potential in Hot Mix Asphalt Using Polymeric Aggregate Treatment

Response 4: The authors thank you for the suggestion. The literature has been expanded with some of the papers recommended.

Point 5: Please use some innovative keywords.

Response 5: Thank you for your suggestion, the keywords have been changed and improved accordingly.

Point 6: Please mention your study limits in the abstract.

Response 6: Thank you for your suggestion. Unfortunately, the limited number of words for the ‘Abstract’ section did not allow us to introduce the study limits. The limits of the present study have been extensively added in the ‘Conclusions’ section as indicated in comment number 2. The authors stated that the present study is a 'preliminary' investigation in order to emphasise the necessary of additional investigations due to study limits.

Point 7: The Conclusions should reflect what the practical application of the results obtained in this study is. In what climatic conditions should the recommendations of the authors be taken into account?

Response 7: The volumetric and mechanical characterizations of the proposed TAO mixes has been performed using the European and Italian technical specifications, hence considering the climatic conditions present in the country of the authors. Based on your suggestion, the authors improved the ‘Conclusions’ section highlighting this aspect. Please, refer to lines 411-413.

Point 8: The authors should increase their discussion on previous related research and highlight how their study is providing a different approach or adding significantly to what has been done. The authors have to explain what is the new here in comparison with the previous studies. The novelty of the current work should be highlighted in the introduction. Please try to mention a problem that needs solving - in other words, the research question underlying your study clearer.

Response 8:  The authors thank you for the useful suggestion. In the literature section, a sentence underling the novelty of the present study was added consisting in using 100% recycled aggregates in bituminous mixtures. Previous works underlined the importance of using sustainable materials in TAO but in different percentages. Please, refer to lines 113-115.

Point 9: The title of the manuscript should be revised.

Response 9: The authors have improved the title of the manuscript following your suggestion.

Point 10: Some types of standards should be used to perform different experimental studies. Please provide details for the standards used in each study.

Response 10: The authors checked and, them, added details on the type of standard used for performing the experimental tests. In general, changes were added to paragraph 2 (Materials) and 3 (Methods), where the standard test methods to characterize the constituent materials and the resulting asphalt mixtures were reported.

Point 11: Section 4 should be discussed in detail.

Response 11: Thank you for all your comments and suggestions about the ‘Conclusions’ section. The paragraph 4 has been improved in terms of discussion details. The authors have tried to better define the topic of the present study, the main findings and the study limits, suggesting future investigations.

Point 12: The authors must redo the Abstract and bring it in compliance with the requirements of the journal. The scientific problem is poorly described (Background). The scientific novelty is not indicated. I recommend shortening the Abstract to 200 words. Editors strongly encourage authors to use the following style of structured abstracts, but without headings: (1) Background: Place the question addressed in a broad context and highlight the purpose of the study; (2) Methods: Briefly describe the main methods or treatments applied; (3) Results: Summarize the article's main findings; and (4) Conclusions: Indicate the main conclusions or interpretations. The abstract should be an objective representation of the article

Response 12: Thank you for your suggestion. According to your comment, the abstract has been improved, always limiting the maximum number of words to 200 as requested by the journal. The abstract was restructured according to reviewer’s suggestions trying to better highlight the background, novelty, methods, results and conclusions sections.

Point 13: It is advisable to add a flowchart at the beginning of the paper. Then the article would become more visual and structured.

Response 13: The authors thank you for the suggestion. A summary of the experimental programme of the research has been added in ‘Introduction’ section. Please refer to Figure 1.

Point 14: The economic aspects are also required for sustainability in social aspect. It is suggested to authors to evaluate the cost-benefit study of this as a further investigation.

Response 14: Thank you for your suggestion. The authors changed the ‘Conclusions’ section accordingly. Please, refer to lines 446-448 for all details.

Point 15: The conclusion should be an objective summary of the most important findings in response to the specific research question or hypothesis. A good conclusion states the principal topic, key arguments and counterpoint, and might suggest future research. It is important to understand the methodological robustness of your study design and report your findings accordingly. Please improve your conclusion section.

Response 15: Thank you for all your comments and suggestions about the ‘Conclusions’ section. The authors have tried to better define the topic of the present study indicating its limits as a preliminary evaluation of the proposed TAO mixes, the main findings, trying to presents them as an objective summary, and the study limits, suggesting future investigations from line 440.

Round 2

Reviewer 3 Report

Accept